# IgG Biomarkers in Multiple Sclerosis: Deciphering Their Puzzling Protein A Connection

**DOI:** 10.3390/biom15030369

**Published:** 2025-03-04

**Authors:** Leonard Apeltsin, Xiaoli Yu

**Affiliations:** 1Anomaly Insights, 500 7th Ave, Floor 8, New York, NY 10028, USA; leonard.apeltsin@gmail.com; 2Department of Neurosurgery, University of Colorado Anschutz Medical Campus, Aurora, CO 80045, USA

**Keywords:** multiple sclerosis, blood biomarker, immunoglobulin, IgG, IgG aggregates, IgM, oligoclonal band, IgG1, IgG3, B cells, IGHV3, IGHV4, complementarity determining region, CDR, framework, protein A, Fc gamma receptor, diagnosis

## Abstract

Identifying reliable biomarkers in peripheral blood is critical for advancing the diagnosis and management of multiple sclerosis (MS), particularly given the invasive nature of cerebrospinal fluid (CSF) sampling. This review explores the role of B cells and immunoglobulins (Igs), particularly IgG and IgM, as biomarkers for MS. B cell oligoclonal bands (OCBs) in the CSF are well-established diagnostic tools, yet peripheral biomarkers remain underdeveloped. Emerging evidence highlights structural and functional variations in immunoglobulin that may correlate with disease activity and progression. A recent novel discovery of blood IgG aggregates in MS patients that fail to bind Protein A reveals promising diagnostic potential and confirms previous findings of the unique features of immunoglobulin G in MS and the potential link between the superantigen Protein A and MS. These aggregates, enriched in IgG1 and IgG3 subclasses, exhibit unique structural properties, including mutations in the framework region 3 (FR3) of *IGHV3* genes, and are associated with complement-dependent neuronal apoptosis. Data based on ELISA have demonstrated that IgG aggregates in plasma can distinguish MS patients from healthy controls and other central nervous system (CNS) disorders with high accuracy and differentiate between disease subtypes. This suggests a role for IgG aggregates as non-invasive biomarkers for MS diagnosis and monitoring.

## 1. Introduction

Multiple sclerosis (MS) is a chronic autoimmune disorder affecting the central nervous system (CNS), marked by inflammation, demyelination, and neuronal damage [1]. Its diagnosis typically relies on clinical evaluations and MRI findings, with cerebrospinal fluid (CSF) analysis often providing valuable support [2]. A key MS biomarker in CSF is the presence of oligoclonal bands (OCBs), composed of immunoglobulin G (IgG), which are detected in about 90–95% of MS patients, signifying an immune response within the CNS and correlating with disease activities [3]. However, obtaining a CSF sample via a lumbar puncture is painful and challenging for patients and healthcare providers. Identifying biomarkers in the blood offers a promising alternative. The success of B cell depletion therapies in managing disease progression while reducing the IgG concentration in the blood indicates that discovering such a biomarker is possible. Our recent reports support the use of blood IgG-based biomarkers for MS and for secondary progressive MS [4]. Building on the unique antibody gene signatures, we present evidence linking MS IgG and the unconventional antigen Staphylococcal Protein A (SPA) and highlight the potential of IgG-based biomarkers in MS diagnosis (Figure 1).

## 2. A Brief Introduction to B Cells and Immunoglobulin Isotypes

B cells originate from hematopoietic stem cells in the bone marrow and generate membrane-bound B cell receptors (BCRs), which are structurally similar to antibodies but attached to the cell surface. Each BCR comprises two heavy and two light chains with complementarity-determining regions (CDRs), enabling antigen binding. While BCRs facilitate direct antigen recognition, antibodies are secreted only after B cell activation [5].

Naïve B cells express IgM, the primary antigen-recognition receptor, and IgD, which regulates activation. Upon T cell-mediated stimulation, they differentiate into antibody-secreting plasma cells or memory B cells, primed for rapid antigen re-exposure. In germinal centers, somatic hypermutation and class-switch recombination refine the antigen affinity and shift antibody production from IgM to IgG, enhancing pathogen neutralization. Plasma cells downregulate CD27 and CD20 while acquiring CD138, marking their transition to dedicated antibody secretion [6].

## 3. The Success of B Cell Depletion Therapies Supports the Biomarker Potential of IgG Antibodies in MS

### 3.1. B Cell Depletion Therapies Target CD20 B Cells and May Reduce IgG Levels in the Blood

B cell depletion therapies such as rituximab and ocrelizumab target CD20-positive B cells, including naïve and memory B cells, in the bloodstream and lymphoid tissues [7]. The current view of this mechanism of action is that these therapies effectively reduce B cell-mediated autoimmune responses but do not directly target plasma cells, which do not express CD20. As a result, immunoglobulin secretion from existing plasma cells continues. However, several recent studies report a reduction in blood IgG levels (Hypogammaglobulinemia) in MS patients after B cell therapies [8,9]. These studies suggest a complex role of B cells and antibodies in MS, and further investigations are needed to clarify the pathophysiology that is induced by B cell therapies.

### 3.2. The Success of B Cell Therapies Demands Better Biomarkers for Early Treatments and Personalized Medicine

B cell depletion disease-modifying therapies represent a high-efficacy treatment option for MS [10]. B cell depletion therapies are administered on a fixed six-month schedule to ensure consistent B cell depletion and prevent relapses. However, this approach does not account for individual variations in disease activity or B cell repopulation, potentially resulting in overtreatment or insufficient suppression. Pre-emptive prediction of MS flare-ups could transform this paradigm. Using biomarkers or other tools to predict relapses, treatments could be timed dynamically and administered only when necessary to suppress immune activity. This personalized approach could optimize therapy, reducing unnecessary immunosuppression while maintaining effective disease control and preserving long-term immune function. Identifying a blood-based biomarker that is linked to the onset and progression of MS could play a pivotal role in making this personalized treatment approach a reality. Serum neurofilament light chain (sNFL) has been intensively investigated as an MS biomarker; however, it is challenging to use it in a clinical setting, as there is a significant overlap between the baseline sNFL level of patients with MS and their controls, who may have migraine or conversion disorder [10,11].

## 4. Antibody Gene Expressions in MS

### 4.1. Antibody Structure Overview

An IgG-based biomarker for MS could leverage immunoglobulin’s gene structure and expression. IgG is a Y-shaped protein with two heavy (H) and two light (L) chains. Each light chain contains a variable (VL) and constant (CL) domain. The heavy chains include a variable (VH) domain followed by three to four constant domains (CH1–CH4). Structurally, IgG comprises two antigen-binding (Fab) regions—each containing VL, CL, VH, and CH1—and a crystallizable (Fc) region (CH2–CH4), forming the molecule’s stem. A flexible hinge between CH1 and CH2 connects Fab and Fc, ensuring structural adaptability [12].

### 4.2. Antibody Classes and Subclasses and Their Relationship to Oligoclonal Bands in MS

Structural variation in the Fc and hinge regions gives rise to different classes (isotypes) of immunoglobulins. These are referred to as IgA, IgD, IgE, IgG, and IgM. The primary structural distinction between IgG and IgM lies in their constant heavy chain domains. IgM contains four constant domains, whereas IgG has only three. Furthermore, IgG can be subdivided into four subclasses (IgG1, IgG2, IgG3, and IgG4) based on the structural variability in the hinge region. In contrast, IgM lacks such subclass differentiation, maintaining a more uniform structure [12]. IgG is the most prevalent and the primary component of oligoclonal bands (OCBs) in the cerebrospinal fluid (CSF), which is utilized for MS diagnosis [3,13,14]; OCBs in MS correlate to disease activities and brain atrophy [3,15,16]. The IgM class, to a lesser extent, can also form OCBs in the CSF of MS patients. Notably, the presence of IgM OCBs is associated with a more aggressive disease course and faster progression, making it a valuable biomarker for identifying high-risk MS cases [17,18]. Thus, current evidence highlights IgG and IgM as the immunoglobulin classes that are most directly associated with MS biomarkers, given their established roles in CSF OCB formation and correlation with disease activity, progression, and prognosis [3,13,14,15,16,17,18]. The remaining Ig classes (IgA, IgD, IgE) are not discussed in this context in the cited literature.

The OCBs in MS are primarily composed of IgG1 [19]. Additionally, IgG3 OCBs are observed in approximately 66% of MS patients, while IgG2 and IgG4 OCBs are much less common [13,20]. The concentration of IgG1 is elevated in the CSF of MS patients [21], and there is a strong correlation between CSF IgG and blood IgG [22]. Furthermore, elevated levels of IgG3 in the peripheral blood (PB) correlate with instances of Clinically Isolated Syndrome (CIS), which is highly suggestive of MS onset [23]. Consequently, detecting differential levels of the IgG1 and IgG3 subclasses within the CSF and peripheral blood of MS patients could provide key insights into their biomarker potentials and improve diagnosis. We showed that higher levels of IgG3 antibodies in serum, but not in CSF, distinguish MS from other neurological disorders [24]. Furthermore, aberrant IgG glycosylation was reported by us, supporting the biomarker potential of blood IgG antibodies [25].

### 4.3. The Biomarker Potential of Antibody Heavy and Light Chain Gene Expression in MS

Further structural distinctions can be made within the heavy and light chains composing the Fab region. The domains within each light chain originate from the immunoglobulin kappa (IGK) locus on chromosome 2 or the immunoglobulin lambda (IGL) locus on chromosome 22. Thus, a light chain can be binarily categorized as belonging to either IGK or IGL. While most light chains are incorporated into intact IG molecules, free light chain (FLC) fragments are not bound to heavy chains. These FLCs circulate freely in the bloodstream and can be found in the CSF. Abnormally elevated free kappa chains (κ-FLCs) are observed in the CSF of most MS patients, which indicates that κ-FLC measurement is a statistically valid alternative to OCB testing [26,27]. However, κ-FLC levels in the PB do not correlate with MS onset or progression [3].

Unlike light chains, which are encoded at two distinct loci on different chromosomes, all heavy chain domains originate from a single locus on chromosome 14. This Immunoglobulin Heavy (IGH) locus contains the constant region C_H_ genes and multiple gene segments that comprise the variable V_H_ domain. The variable domain is formed through the recombination of three distinct gene segments: *Variable* (V), *Diversity* (D), and *Joining* (J). Each gene segment is associated with multiple gene families. For instance, seven unique V families are located on chromosome 14. These families are designated as *IGHV1* through *IGHV7*. The *IGHV* families are further divided into functional genes, varying in diversity by family [28]. For example, the *IGHV2* family includes only two functional genes, *IGHV2-5* and *IGHV2-70*. In contrast, the *IGHV3* family contains approximately 20 functional genes, with *IGHV3-23* being the most frequently expressed [29].

Studies have shown that specific genes from the *IGHV4* family, particularly *IGHV4-34* and *IGHV4-39*, are associated with MS. These genes are overrepresented in clonally expanded B cells within the CSF of MS patients compared to those with other neurological diseases (OND: chronic meningitis of unknown etiology and Subacute sclerosing panencephalitis) [30,31,32,33]. Additionally, an increased representation of *IGHV4-59/61* and *IGHV4-4* genes in the CSF of MS patients was found [32]. However, this preferential usage of IGHV4 family genes in the CSF does not extend to the blood. Instead, in the peripheral blood mononuclear cells (PBMCs) of MS patients, the *IGHV3-23* and *IGHV3-74* subfamilies are overrepresented when compared to PBMCs from OND patients. Nonetheless, this overrepresentation is insufficient to allow for statistically precise biomarker development [32,34]. Using mass spectrometry proteomics, we discovered that the IgG heavy chains *IGHV1–3*, *IGHV3–43*, and *IGHV3–49* were present in the IgG aggregates for relapsing–remitting MS (RRMS) and secondary progressive MS (SPMS) but not in controls [35].

It should be noted that light chains on the IGK and IGL loci also undergo recombination across various variable gene families. While most of these light chain genes have not been correlated to MS, there is an exception for *IGLV3-21*, which shows higher expression in the PB and CSF of MS [34,36]. However, these observed differences are insufficient for precise diagnostics.

## 5. The Unique MS-Specific Motifs in the Antigen-Binding Variable Domain of Antibodies

### 5.1. The Heavy Chain CDR3 Plays a Crucial Role in Determining Antigen-Binding Characteristics

The antigen-binding V domain of antibodies exhibits significantly greater sequence variation and immune response specificity compared to other antibody regions. Thus, subtle mutations in this V domain could help reveal diagnostically relevant features that are specific to MS. The V domain contains three non-consecutive CDRs: CDR1, CDR2, and CDR3. These CDRs are flanked by four framework regions (FRs), which provide structural support to ensure that the CDRs can appropriately interact with antigens. The FR regions are highly conserved due to their structural importance, while the CDRs are highly variable, allowing for precise tuning towards specific antigen interactions. Among the CDRs, CDR3 is notable for its exceptional diversity [5]. While CDR1 and CDR2 are located exclusively on the variable gene segment, the CDR3 region is a product of gene recombination. The light chain CDR3 (CDR-L3) is constructed by recombining the *V* and *J* gene segments. The heavy chain CDR3 (CDR-H3) is formed through the more combinatorially complex V(D)J recombination. Among the six CDRs that are spread across heavy and light chains, CDR-H3 exhibits the highest variability, and its diversity suggests a central role in determining antigen-binding specificity [5].

### 5.2. CDR-H3 as Biomarkers for MS and Other Autoimmune Disorders

Disease-specific patterns in the CDR-H3 region have been identified in other autoimmune disorders. For instance, arginine-rich CDR-H3 motifs have been linked to lupus pathogenesis [37]. Additionally, anomalously short CDR-H3 lengths have been observed in patients suffering from neuromyelitis optica spectrum disorder (NMOSD) [38]. These anomalies are associated with a unique antibody, NMO-IgG, which serves as a blood-based biomarker for NMOSD [39]. This finding is particularly relevant to MS biomarker research, since NMOSD is also a demyelinating autoimmune disease targeting the CNS, like MS. In fact, due to symptom overlap, approximately 40% of NMOSD patients are misdiagnosed with MS or other diseases [40]. Consequently, it is plausible that CDR-H3 analysis could eventually uncover a comparable biomarker in the peripheral blood of MS patients.

Researchers have utilized next-generation sequencing technologies to analyze millions of unique CDR-H3 sequences from the CSF and PB of MS patients and controls [32,36,41]. These studies demonstrated that approximately one-third of B cells in MS CSF are clonally related to peripheral B cells, shedding light on B cell movement across the blood–brain barrier in the diseases. However, despite these efforts, studies have failed to identify a CDR-H3 motif that correlates with MS diagnosis. Consequently, the quest to discover a unique disease-driving antibody akin to NMO-IgG in NMOSD has proven elusive.

### 5.3. MS-Specific IGHV3 Gene Mutation in Framework 3: An Indication of the Role of Superantigen Protein A

Expanding sequence analysis beyond the CDR-H3 region to include the entire heavy chain variable domain has proven more productive. In a 2017 study involving 97 individuals (51 MS patients and 46 healthy controls), researchers identified an MS-specific V domain motif within the PBMC repertoires [42]. Statistical analysis revealed a shared insertion–deletion mutation in the *IGHV3* gene that was present in 35 MS patients but absent in all the healthy controls. This mutation caused a localized reading frame shift within framework region 3 (FR3) of *IGHV3*. Typically, framework regions do not directly participate in antigen interactions, except in the case of certain superantigens that bind outside the CDRs of their target receptors [43]. For instance, the FR3 region, in conjunction with CDR-H2, is known to contribute to the specific affinity of the *IGHV3* family for the superantigen Staphylococcal Protein A (SPA) [44]. Modifications to the FR3 region can significantly disrupt Protein A binding [44,45]. Therefore, this mutated MS-specific motif could be inferred by assessing Protein A binding activity. Table 1 summarizes the terms used for this review.

## 6. Protein A Binding to IgG and Its Critical Role in Multiple Sclerosis

### 6.1. The Unique Feature of Protein A Binding to IgG in MS

Protein A binds with high affinity to the Fc region of immunoglobulins [46] and has been extensively used for the purification of IgGs [47]. The unexpected association between superantigen Protein A’s high affinity for framework 3 and the detection of toxic IgG aggregates in Protein A flow-through samples from MS patients suggests a potential link to the disease. This highlights the potential role of antibody-binding affinity in the MS pathology. We recently showed that IgG antibodies in MS blood form large aggregates (>100 nm) that do not bind efficiently to Protein A [35]. These aggregates, retained in the flow-through after Protein A binding, support the structural changes in MS IgG that hinder typical interaction with the superantigen Protein A. This report aligns with the gene analysis study of FR3 mutation in *IGHV3*, which disrupts Protein A binding, providing additional proof of the importance of IgG in the MS disease pathogenesis and the strong connection between Protein A and MS. Therefore, IgG aggregates mediate and complement neuronal cytotoxicity, supported by additional gene studies, which points to their potential biomarker role in MS. It is not surprising that our follow-up studies revealed that the highly elevated levels of IgG1, IgG3, and total IgG in the IgG aggregates of MS blood serve as biomarkers for MS and for secondary progressive MS [4]. The IgG aggregates function as MS biomarkers and therefore demonstrate the biological rationale for IgG gene mutation and IgG-mediated neuronal cytotoxicity, supporting the idea that MS is a B cell-mediated immune dysregulated disease.

### 6.2. The Enriched IgG Aggregates in Protein A Flow-Through Cause Neuron Cytotoxicity and Can Be Used as Biomarkers for MS

Using ELISA, we demonstrated that plasma IgG levels within these aggregates can effectively distinguish MS patients from healthy controls [4]. We included two cohorts (190 MS patients and 160 controls). We collected the IgG aggregates, enriched in the flow-through after plasma exposure to Protein A, followed by the detection of IgG subclasses. Significantly higher levels of IgG1, IgG3, and total IgG were found in the IgG aggregates of MS patients, achieving an area under the curve (AUC) greater than 90% in diagnostic tests. Moreover, elevated levels of IgG1 were able to differentiate SPMS from RRMS, with an AUC of 91%. The strong correlation between these IgG antibodies and IgG aggregate-induced neuronal cytotoxicity provides a biological rationale for their use as plasma biomarkers [4]. These findings support the use of less-invasive, simple IgG-based blood ELISA assays in clinical practice for diagnosing MS, distinguishing disease subtypes, and monitoring treatment responses, thereby reinforcing the potential of PBMC-derived biomarkers in managing this complex autoimmune disease. A recent report that the autoantibody signature can predict MS [48] supports the biomarker potential of blood IgG.

## 7. Speculative Mechanisms of IgG Aggregate Formation in MS and Neuronal Apoptosis

Unanswered questions remain regarding the biological mechanisms behind the proposed IgG aggregates in the Protein A flow-through as biomarkers for MS. For instance, it is unknown how the lack of binding to Protein A correlates with aggregate formation. To explore this question, we can examine the crystallized structure of the interaction between *IGHV3* and Protein A (Figure 2). In this structure, Protein A binds to the antibody’s Fc and Fab regions. Notably, the FR3-inclusive site that is responsible for Fab binding is structurally separate from the domain surface that mediates Fc binding [45]. Based on the previously observed MS-specific FR3 indel mutation, we speculate that the non-Protein A-binding flow-through aggregates contain mutations either in the FR3 region of the Fab or in both the Fab and Fc regions. Either scenario could explain the formation of these aggregates. Mutations in the FR3 region can destabilize the structural support of the variable domain, leading to partial unfolding and subsequently causing aggregative interactions along the misfolded regions of the antibody [49,50]. Additionally, mutations in the Fc region can further facilitate aggregation through cross-antibody Fc-Fc interactions [51].

Further questions remain regarding how the IgG aggregates trigger neuronal apoptosis. One plausible explanation involves the engagement of Fc gamma receptors (FcγRs), which bind to the Fc portion of IgG antibodies. Aberrant activation of FcγRs on neural cells may contribute to the pathogenesis of primary neurodegenerative conditions, including Alzheimer’s disease, Parkinson’s disease, ischemic stroke, and MS [52]. Although the binding affinity between a single IgG molecule and an FcγR is relatively low, the formation of IgG aggregates allows for multivalent binding [53]. This means that multiple IgG molecules within an aggregate can simultaneously engage several FcγRs on the neuronal cell surface, significantly enhancing the overall binding strength. Such aberrant receptor interactions could initiate intracellular signaling cascades that promote inflammation and apoptosis, potentially leading to neuronal damage. Additionally, it was shown recently that Protein A inhibits complement activation by interfering with IgG hexamer formation [54].

## 8. Conclusions

The quest for peripheral blood biomarkers in MS is not only a scientific imperative but also a clinical necessity, given the invasive nature of cerebrospinal fluid sampling [55]. Meanwhile, traditional approaches focusing on specific immunoglobulin gene motifs and antigen specificities have not yielded definitive diagnostic tools [19,56]. Emerging evidence on IgG aggregates that do not bind to Protein A offers promising new directions. These aggregates, enriched in IgG1 and IgG3 subclasses, have demonstrated high diagnostic accuracy and are linked to neuronal apoptosis [35], bridging peripheral immune responses with central nervous system degeneration. Understanding the mechanisms behind these aggregates could unlock new pathways for disease monitoring and personalized treatment strategies.

## Figures and Tables

**Figure 1 biomolecules-15-00369-f001:**
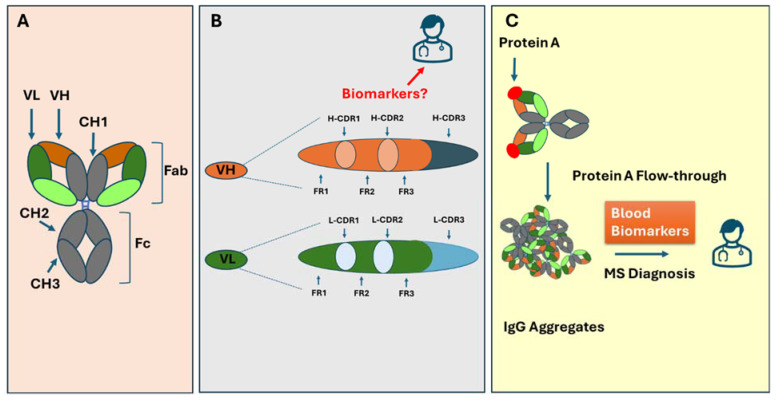
Graphic overview depicting the antibody structure, the arrangement of CDRs and their framework in heavy and light chains, and the binding of Protein A, resulting in the enrichment of IgG aggregates. (**A**) An antibody is composed of a heavy and light chain with Fab and Fc fragments. (**B**) CDR1-CDR3 and the framework segments in heavy and light chains. (**C**) The binding of Protein A resulted in the enrichment of IgG aggregates, which can be used as MS biomarkers.

**Figure 2 biomolecules-15-00369-f002:**
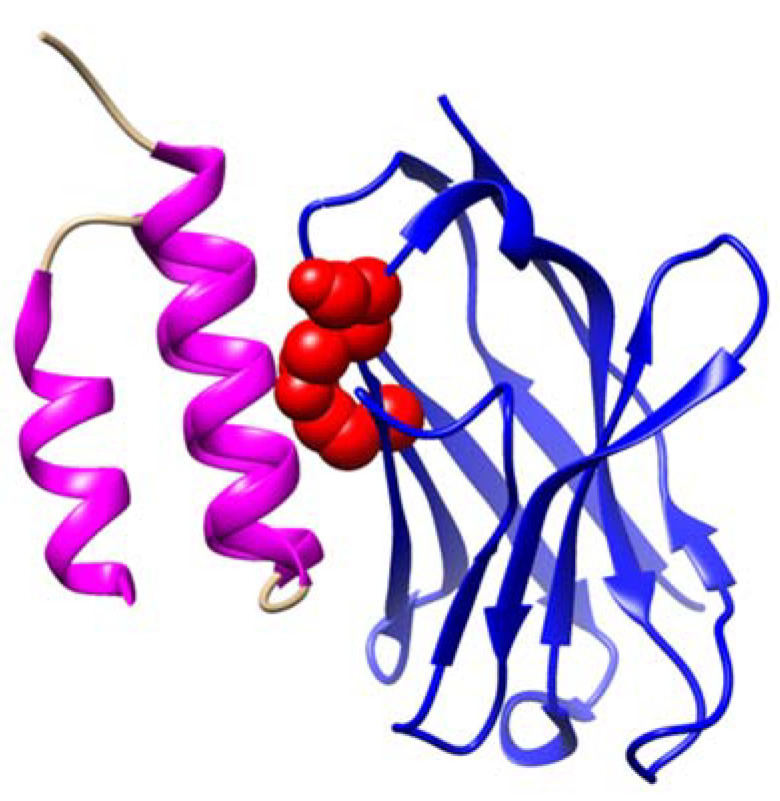
Structure of a Protein A domain (magenta) binding to a germline *IGHV3* variable chain (blue). Three FR3 residues (Gln-H81, Asn-H82a, Ser-H82b) that mediate the binding are highlighted as red atomic spheres. These residues are impacted by indel mutation, observed exclusively in MS patients, which disrupts Protein A binding. Here, Protein A is shown to bind to the region near the CDR, whereas in most figures, it is bound to CH2.

**Table 1 biomolecules-15-00369-t001:** An overview of key antibody structural components and their potential value as biomarkers for MS. Notably, the biomarker signals found in the PB are often linked to how these antibody components interact with Protein A.

Name (Acronym)	Description	CSF Biomarker for MS	Blood Biomarker for MS
Immunoglobulin M **(IgM)**	It is expressed by naïve B cells and contains four constant domains.	Found in OCBs, which are associated with disease progression.	N/A
Immunoglobulin G **(IgG)**	IgG is expressed after class-switch recombination, which contains three constant domains.	The most common and abundant Ig class in OCBs [13].	N/A
Immunoglobulin G1 **(IgG1)**	IgG class with a short hinge region.	Primary component of OCBs. IgG1 is elevated in MS CSF [19].	Elevated levels of IgG1 were found in the IgG aggregates in MS [4].
Immunoglobulin G3 **(IgG3)**	IgG3 has a longer, more flexible hinge region compared to IgG1.	IgG3 OCBs are observed in approximately 66% of MS patients [13].	Elevated levels of IgG3 were found in the IgG aggregates in MS [4,23].
Fragment Antigen-Binding Region **(Fab)**	Region of the antibody that binds to antigens, composed of a heavy chain and a light chain.	N/A	N/A
Immunoglobulin Kappa Light Chain **(IGK)**	Light chain originating from the Ig Kappa locus on chromosome 2, which is composed of one variable domain and one constant domain.	Abnormally elevated free kappa chains are observed in the CSF of most MS patients [26,27].	N/A
Immunoglobulin Heavy Chain **(IGH)**	The heavy chain is composed of one variable domain and three to four constant domains. The variable domain is formed through the recombination of three distinct gene segments: V, D, and J.	N/A	N/A
*Immunoglobulin Heavy Variable Gene* * **(IGHV)** *	Seven unique V families are located on chromosome 14.	N/A	N/A
*Immunoglobulin Heavy Variable 4 Gene* * **(IGHV4)** *	One of the seven *IGHV* families.	*IGHV4-34* and *IGHV4-39* are overrepresented within the CSF of MS patients [32,36].	N/A
Immunoglobulin Heavy Variable 3 Gene **(*IGHV3*)**	One of the seven *IGHV* families.	N/A	The *IGHV3-23* and *IGHV3-74* subfamilies are overrepresented in MS [29].
Heavy Chain Complementarity Determining Region 3 **(CDR3-H)**	The most diverse CDR region is produced through VDJ gene recombination [5].	N/A	Disease-specific patterns in CDR3-H have been identified in other autoimmune diseases but not in MS [37,38].
*IGHV3* Framework Region 3 **(*IGHV3* FR3)**	The region between CDR2-H and CDR3-H uniquely interacts with Protein A [44,45].	N/A	A distinct indel mutation in the *IGHV3* FR3 region was observed in MS blood [42].

## Data Availability

No new data were created or analyzed in this study.

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
