# Peer review of "IgG Biomarkers in Multiple Sclerosis: Deciphering Their Puzzling Protein A Connection"

_biomolecules, 2025, doi:10.3390/biom15030369_

Round 1
Reviewer 1 Report
Comments and Suggestions for Authors
The manuscript needs to incorporate the reference of every article mentioned in Table 1 (those related to IGHV3 and CDR3-H are missing).
Author Response
The manuscript needs to incorporate the reference of every article mentioned in Table 1 (those related to IGHV3 and CDR3-H are missing).
Response. We appreciate the suggestions. We added appropriate references to the table in the revised manuscript.
Reviewer 2 Report
Comments and Suggestions for Authors
See the attachment, please.

Author Response
- In my opinion, the manuscript does not fully reflect the idea of a typical review. The large part of the paper is a description of authors’ earlier work or a reference to that work (see sections 1., 4.2., 4.3., 5.2.). 15 out of 57 (26%) references are autoreferences. I feel this is more of a personal summary than a review.
Response: We appreciate your opinion. IgG antibodies and oligoclonal bands have been the critical research area for decades. However, due to the challenge of the investigations and the dwindling funding support, few groups continue the studies in the area, resulting in less publications by others. We are proud to cite our papers in the current review. Nonetheless, we have replaced several citations in Sections 1, 4.2, and 4.3 with alternative, equally important references to reduce auto-referencing. As a result, only 17% of the 57 citations now refer to previous work by the authors.
Regarding the studies on the relationship between Protein A and MS biomarker development, Dr. Yu and Dr. Apeltsin conducted their research independently at different universities and published their findings seven years apart, without prior knowledge of each other’s work. These references do not reflect collaborative research but rather independent contributions to the field. Given their relevance, we believe their inclusion is justified as part of a broader scientific review on biomarker development
- Some sections contain very basic knowledge (such as 2. and 4.1.) and they could be shortened or moved to the beginning of the manuscript. Some ideas are introduced but explained further in the text. For example, protein A is firstly mentioned in the introduction but explained in the section 6.1., and in the section 5.3. suddenly SPA abbreviation is introduced.
Response. Thank you for the insight. Subsections 2.1 and 2.2 have been condensed into a single paragraph to streamline the presentation. Similarly, Section 4.1 has been significantly shortened; however, we have retained its placement to ensure continuity with subsequent subclass discussions, as well as to ensure that the review’s ontological resolution flow from large-scale (B-cell types) to smaller scale (antibody categories followed by individual antibody sub-components). To improve clarity, Staphylococcal Protein A and its abbreviation (SPA) are now introduced more explicitly in the introduction. However, the detailed discussion of Protein A remains in later sections, as it relies on prior explanations of motif detection and framework regions. This structure ensures a logical progression of ideas. Additionally, keeping the in-depth discussion of Protein A toward the end aligns with its initial mention in the introduction, which provides an ordered outline of key topics covered in the paper.
- Section 4.1. contains description of immunoglobulin protein structure, however the title suggests gene structure description.
Response: Section 4.1 has been renamed to “Antibody gene structure overview.”
- There are some examples of claims not supported by sources: 137, 159-160, 206
Response: We have revised the text to ensure all claims are explicitly supported by the cited references. The language has been adjusted to align factual statements with the evidence provided in the sources. Overreaching conclusions have been qualified, and unsupported assertions have been removed or contextualized.
- Editing and abbreviation/name unification could be improved in the manuscript. There are numerous unexplained abbreviations used throughout the whole manuscript (such as RRMS, SPMS) and abbreviations not explained the first time they appear in the text (such as CNS, CDR). Some abbreviations are unnecessary, such as NGS, AUC, or HCs and OND copied from the reference papers used simply as a research group designations. Some abbreviations are explained several times in the manuscript (for example in lines 165 and 203).
Response: Abbreviations have been revised in accordance with this feedback. Those first introduced in the abstract are now repeated in the introduction for clarity. However, we have retained the abbreviation for Other Neurological Diseases (OND), as it is used twice within a later sentence referring to 'OND patients.' Without this abbreviation, the sentence would become either overly long or unclear. Additionally, the plural abbreviation 'CDRs' is maintained as a suitable substitute for the singular 'CDR' where appropriate.
- Use either B-cells or B cells in the manuscript.
Response: B cells are now used consistently in manuscript.
- Use either CD20 or CD 20 in the manuscript.
Response: CD20 now used consistently.
- Use “mass spectrometry” instead of Mass Spec.
Response: Addressed.
- There are some random capital letters in the middle of sentences.
Response: Capitalization has been appropriately adjusted.
- Change “MS” to “MS patients” in the sentences where you are describing patients (example: line 220).
Response: Addressed
- Write “CDR-H3 as Biomarkers for MS and Other Autoimmune Disorders” in the line 207.
Response: Addressed.
- Reference style does not match MDPI’s style. Use [1] instead of (1).
Response: Revised.
13 Line 68 – do you mean “antigen/antigens” by writing “antigen (s)”?
Response: Edited during section revision.
- Some sentences are long and/ or hard to understand, for example in lines 59-61, 170-171, 247- 251.
Response: Sentences have been edited for clarity. Overly long statements have been broken up into multiple sentences.
- Do not use “non-invasive” in context of blood collection. Blood collection is an invasive procedure. You may describe it a less invasive than cerebrospinal fluid collection.
Response: Switched to “less-invasive”.
- There are many double spaces.
Response: We did our best to make the corrections.
- Figure 1B. For clarity, label brown ellipse as VH and green ellipse as VL.
Response: Revised.
- Figure 2. Is the figure authors’ own work? It could be smaller.
Response: Yes. We made the figure smaller.
The paper does not contain mandatory sections such as “funding” or “author contributions
Response: Added in the revised manuscript.
Reviewer 3 Report
Comments and Suggestions for Authors
Fine review. I only have minor comments.
Line
15 immunoglobulins
18 and a potential link (there is nothing mysterious about nature !)
52 the cells responsible for
96 depleting
113 allow to
180 Spectrometry
186 MS patients
214 research,
245 Protein A
250 a potential link (there is nothing mysterious about nature !)
250 revealing that
270 plasma exposure to
271 IgG were
287 this question, we examined (there is nothing mysterious about nature !)
291 we speculate
309 means that
302 disrupts Protein
321 offers a promising new direction.
Fig. 1 Protein A is shown bound to the CDR, whereas in most figures, it is bound to CH2
Author Response
Response: Thank you for the minor comments and suggestions. The manuscript has been updated to incorporate all the requested changes.
Round 2
Reviewer 2 Report
Comments and Suggestions for Authors
See the attachment, please.

See the attachment, please.
Author Response
Dear Reviewer, Thank you for your critical review and insightful comments. We appreciate your time and effort in making this manuscript a better paper. Based on your critiques, we revised the manuscript point by point. The newly revised words/sentences are highlighted in purple.
Thanks again!
Dr. Y. and Dr. A.